# Amphibian (*Xenopus laevis*) Macrophage Subsets Vary in Their Responses to the Chytrid Fungus *Batrachochytrium dendrobatidis*

**DOI:** 10.3390/jof11040311

**Published:** 2025-04-15

**Authors:** Amulya Yaparla, Milan Popovic, Kelsey A. Hauser, Louise A. Rollins-Smith, Leon Grayfer

**Affiliations:** 1Department of Biological Sciences, George Washington University, Washington, DC 20052, USA; ayaparla@arcellx.com (A.Y.); grudvica@gmail.com (M.P.); kelsey.hauser@noblis.org (K.A.H.); 2Departments of Pathology, Microbiology and Immunology, and of Pediatrics, Vanderbilt University School of Medicine, Nashville, TN 37232, USA; louise.rollins-smith@vanderbilt.edu; 3Department of Biological Sciences, Vanderbilt University, Nashville, TN 37232, USA

**Keywords:** *Batrachochytrium dendrobatidis*, chytridiomycosis, macrophage, antifungal, myelopoiesis, amphibian immune response, CSF1, IL34

## Abstract

The chytrid fungus, *Batrachochytrium dendrobatidis* (Bd), infects amphibian skin, causing chytridiomycosis, which is a contributing cause of worldwide declines and extinctions of amphibians. Relatively little is known about the roles of amphibian skin-resident immune cells, such as macrophages, in these antifungal defenses. Across vertebrates, macrophage differentiation is controlled through the activation of colony-stimulating factor-1 (CSF1) receptor by CSF1 and interleukin-34 (IL34) cytokines. While the precise roles of these respective cytokines in macrophage development remain to be fully explored, our ongoing studies indicate that frog (*Xenopus laevis*) macrophages differentiated by recombinant forms of CSF1 and IL34 are functionally distinct. Accordingly, we explored the roles of *X. laevis* CSF1- and IL34-macrophages in anti-Bd defenses. Enriching cutaneous IL34-macrophages, but not CSF1-macrophages, resulted in significant anti-Bd protection. In vitro analysis of frog macrophage-Bd interactions indicated that both macrophage subsets phagocytosed Bd. However, IL34-macrophages cocultured with Bd exhibited greater pro-inflammatory gene expression, whereas CSF1-macrophages cocultured with Bd showed greater immunosuppressive gene expression profiles. Concurrently, Bd-cocultured with CSF1-macrophages, but not IL34-macrophages, possessed elevated expression of genes associated with immune evasion. This work marks a step forward in our understanding of the roles of frog macrophage subsets in antifungal defenses.

## 1. Introduction

The chytrid fungus *Batrachochytrium dendrobatidis* (Bd) is a leading cause of worldwide amphibian declines and extinctions [1,2,3,4,5]. The swimming zoospore stage colonizes the keratinized skin of adult amphibians and the keratinized mouthparts of tadpoles [1], forming germination tubes [6,7,8], and sending their contents into epidermal cells [7,8] to mature. The invading zoospores develop into urn-shaped zoosporangia in which new zoospores develop [1,2,3] and swim out to infect other areas of the skin or new hosts [6]. Bd infections remain confined to the amphibian skin [1,2,3,6], causing death by interfering with the essential ion transport across the skin, thereby leading to cardiac arrest [9,10,11].

Fungal pathogens typically have complex interactions with their hosts, and while host immune defenses may keep these pathogens at bay [12], many fungi have evolved elaborate immune evasion strategies [13,14,15]. Bd is likewise believed to be capable of substantial immune evasion [1,3,16]. In turn, considerable research has shown that both the innate and adaptive arms of amphibian immunity are engaged during Bd infections [16,17,18,19,20,21,22,23,24,25,26,27,28,29]. Some notable hallmarks of amphibian anti-Bd defenses include antifungal products from symbiotic bacteria [27,30,31,32,33,34,35,36]; mucus and granular gland secretions containing antimicrobial peptides [16,23,28,29,37,38,39,40]; alkaloids [41,42,43]; and lysozymes [44,45]; as well as antibodies [33]. However, we know almost nothing about the contribution of skin-resident sentinel immune cells, such as macrophages (Mϕs), to anti-Bd defenses.

Mϕ-lineage immune cells are important to vertebrate immune responses, playing key roles in tissue surveillance, pathogen detection, and coordinating ensuing immune responses [46,47]. In mammals, Mϕ-lineage cells known as Langerhans cells are critical to immune surveillance of skin epithelia [47], and Langerhans-like cells have been described in amphibians [48]. Relatively little is known about amphibian Mϕ-lineage populations, and reagents to distinguish amphibian immune subsets corresponding to mammalian Mϕ subtypes, such as Langerhans cells, are lacking. While numerous distinct Mϕ-lineage subsets have been described [49,50], the differentiation and functionality of all vertebrate Mϕs depend on colony-stimulating factor-1 receptor (CSF-1R), which is ligated by CSF-1 and interleukin-34 (IL34) cytokines [51]. IL34, but not CSF1, additionally ligates protein-tyrosine phosphatase zeta (PTPζ) and CD138 (Syndecan-1, reviewed in [52]). A growing body of literature indicates that Mϕs differentiated by CSF1 and IL34 possess non-overlapping phenotypes and roles (reviewed in [52,53]). Our studies in the *Xenopus laevis* frog likewise indicate that Mϕs differentiated by recombinant (r)CSF-1 and rIL-34 possess markedly distinct capacities to recognize and respond to disparate pathogens [54,55]. Moreover, *X. laevis* possess a substantial expression of both *csf1* and *il34* genes in their skin [56], indicating that both CSF-1- and IL-34-Mϕs reside therein.

In this study, we investigated the distinct roles of *Xenopus laevis* CSF1- and IL34-differentiated Mϕs during *Bd* infection, using both in vivo and in vitro approaches to assess their contribution to antifungal defense and host–pathogen interactions.

## 2. Materials and Methods

### 2.1. Animals

Adult *Xenopus laevis* (~1–2-year old) were purchased from Xenopus1 (Dexter, MI, USA). The animals were housed and handled under strict laboratory regulations of the Animal Research Facility at the George Washington University (GWU) per the GWU Institutional Animal Care and Use Committee regulations (approval number A2023-046).

### 2.2. Batrachochytrium Dendrobatidis JEL197

The Bd isolate JEL197 was cultured in 1% tryptone broth, supplemented with 100 U/mL penicillin and 100 μg/mL streptomycin (Gibco, Waltham, MA, USA) at 20 °C for 3–7 days. For in vivo infection studies, Bd cultures were filtered through sterile coffee filters to isolate and enumerate zoospores. For Bd-Mϕ coculture, Bd thalli were enumerated by the trypan blue exclusion method and cocultured at ratios of 5 Bd thalli per Mϕ.

### 2.3. Recombinant Cytokine Production

Recombinant (r)CSF1 and rIL34 were produced using previously described approaches [55]. Briefly, signal-peptide-cleaved transcripts corresponding to *X. laevis* CSF1 or IL34 were PCR-amplified and ligated into the pMIB/V5 His A insect expression vector (Invitrogen, Waltham, MA, USA) and used to transfect Sf9 insect cells with the transfection reagent cellfectin II (Invitrogen). Western blot analysis against the V5 epitope was performed on the supernatants from transfected Sf9 cells to confirm the expression of rCSF1 or rIL34, following the selection of positive transfectants using 10 μg/mL blasticidin. The cultures were then scaled up to 500 mL and grown until confluent (5–7 million cells/mL). The confluent cultures were pelleted to obtain supernatants, which were concentrated against polyethylene glycol flakes (8 kDa) at 4 °C to 100 mL volume. The concentrated supernatants were dialyzed overnight at 4 °C against 150 mM sodium phosphate, pH 8. Recombinant cytokines were isolated from the dialyzed supernatants via Ni-NTA agarose columns (Qiagen, Germantown, MD, USA). Bound proteins were washed six times with low-stringency wash buffer (0.5% Tween 20, 50 nM sodium phosphate, 500 mM sodium chloride, 40 mM imidazole). The recombinant CSF1 and IL34 were eluted in fractions with the elution buffer (as above, but with 250 mM imidazole), which were pooled and stored as aliquots at −20 °C. Protein concentrations were quantified by BCA (ThermoFisher, Waltham, MA, USA) assays. The purity of the recombinants was tested by Coomassie stains of proteins resolved on SDS-gels and Western blot against the V5 epitope (Appendix A).

The recombinant control (rctrl) was generated by transfecting Sf9 cells with an empty pMIB/V5 His A insect expression vector (Invitrogen), selecting positive transfectants with blasticidin, scaling up these cultures and processing the derived supernatants using the same approaches described above for rCSF1 and rIL34 production.

### 2.4. Subcutaneous Administration of rCSF1 or rIL34 and In Vivo Bd Challenge

To enrich cutaneous frog Mϕs, the adult *X. laevis* were injected subcutaneously once between the dorsal skin and underlining muscle layer with rCSF1, rIL34 (5 μg in 20 μL APBS per animal), or equal volumes of the rctrl (*N* = 4 per treatment group). After 3 days, the animals were sacrificed by MS-222 overdose followed by decapitation, and their skins were isolated and fixed in 10% neutral buffered formalin. The skins were processed and embedded in paraffin, sectioned (5 μm thickness) by the GWU Pathology Core. The sections were then stained with α-Naphthyl Acetate (Non-Specific Esterase; Sigma, Darmstadt, Germany), according to the manufacturer’s instructions, and optimized to frog tissues (diluted hematoxylin solution 1/3 in APBS) to visualize cutaneous Mϕs.

After confirming Mϕ enrichment quantitatively, the frogs were again injected dorsally with rCSF1, rIL34 (5 μg in 20 μL APBS/animal, based on prior work deriving peritoneal Mϕs [56]) or equal volumes of the rctrl (*N* = 7 per treatment group, based on prior experiments of a similar nature), and after 3 days, the animals were challenged with Bd by water bath (10^7^ Bd zoospores in 500 mL water). After 7 days of Bd-exposure (without water change), the animals were sacrificed, and their dorsal skins were excised and flash-frozen over dry ice in Trizol (Invitrogen) for later DNA isolation.

### 2.5. Bone Marrow-Derived Mϕ Cultures and In Vitro Mϕ-Bd Challenge

Amphibian serum-free (ASF) [57] medium supplemented with 10% fetal bovine serum, 0.25% *X. laevis* serum, 10 μg/mL gentamycin (Thermo Fisher Scientific, Waltham, MA, USA), 100 U/mL penicillin (Gibco), and 100 μg/mL streptomycin (Gibco) was used for all Mϕ cultures. Amphibian phosphate-buffered saline (APBS) has been described [37].

Frog bone-marrow-derived CSF1- and IL34-Mϕ cultures were established using previously described methods [55]. Briefly, femur bones from the adult *X. laevis* were flushed with saline, and the bone marrow cells were enumerated by the trypan blue exclusion method and seeded at a density of 10,000 cells/well in a 96-well culture plate. Cells were treated with 250 ng/mL of either recombinant CSF1 or IL34 to generate CSF1- or IL34-Mϕs, respectively, and incubated at 27 °C with 5% CO_2_. After 5 days of culture [58], the cells were challenged with developing Bd thalli (5 fungal cells per Mϕ) or mock challenged.

To assess Mϕ responses when in coculture with Bd, CSF1- or IL34-Mϕ cultures, generated as above, were incubated alone or with Bd (5 Bd cells/Mϕ, based on preliminary data) for 5 h before being collected for microscopy, RNA isolation (Trizol, Invitrogen), cDNA synthesis, and quantitative gene expression analyses.

### 2.6. Quantitative Gene Expression and Bd Load Analyses

The total RNA from the frog skin tissues, in vitro Mϕ cultures, and Mϕ-Bd cocultures were isolated using a Trizol RNA extraction protocol. For all experiments, the tissues and cells were collected into Trizol reagent (Invitrogen) and homogenized by passage through progressively higher gage needles. After homogenization, 500 ng of total RNA was used for cDNA synthesis using cDNA qscript supermix (Quantabio, Beverly, MA, USA), according to the manufacturer’s instructions. The CFX96 Real-Time System and iTaq Universal SYBR green supermix (Quantabio) were used for qPCR analyses using the delta^delta CT method. All gene expression was examined relative to the glyceraldehyde-3-phosphate dehydrogenase (*gapdh*) endogenous control. All primers were validated prior to use. All primer sequences are provided in Appendix A.

DNA was isolated from the Trizol following RNA isolation. In brief, following phase separation and RNA extraction, back-extraction buffer (4 M guanidine thiocyanate, 50 mM sodium citrate, 1 M Tris pH 8.0) was added to the remaining Trizol layers, and the mixtures were centrifuged to isolate the DNA-containing aqueous phase. The DNA was precipitated overnight against isopropanol at −20 °C, pelleted by centrifugation, washed with 70% ethanol, and resuspended in TE buffer (10 mM Tris pH 8.0, 1 mM EDTA). The DNA was further purified by phenol–chloroform extraction and resuspended in molecular-grade water.

Bd loads were quantified as previously described [59]. In brief, a standard curve was prepared by serial dilutions of Bd (JEL 197) zoospore DNA. This standard curve was used in absolute qPCR reactions, with 12 μg total input DNA per sample and using previously validated primers against the Bd ribosomal RNA internal transcribed spacer 1 (ITS1; Appendix A). All qPCR assays were performed using iTaq Universal SYBR Green Supermix (Bio-Rad Laboratories, Hercules, CA, USA), and all experiments were analyzed via a CFX96 Real-Time System (Bio-Rad Laboratories, Hercules, CA, USA) and BioRad CFX Manager software (SDS, V 3.1).

### 2.7. Electron Microscopy

Electron microscopy was performed to visualize differences between the two Mϕ subsets with respect to the morphology and phagocytosis of Bd. The processing and imaging of the cells by transmission and scanning electron microscopy (TEM and SEM) was performed at the GWU Nanofabrication and Imaging Center (GWNIC). For TEM, the Mϕ or Mϕ-Bd cocultures were fixed as monolayers on six-well plates with 2.5% glutaraldehyde and 1% paraformaldehyde in 0.1 M sodium cacodylate buffer for one hour. The cells were treated with 1% osmium tetroxide in 0.1 M sodium cacodylate buffer for 1 h. Following washes, the cells were *en bloc*-stained with 1% uranyl acetate in water overnight at 4 °C. The samples were dehydrated via an ethanol series and embedded in epoxy resin using LX112. Inverted Beem capsules were placed into each tissue culture well to create *on face* block-faces for sectioning. Resin was cured for 48 h at 60 °C. The 95 nm sections were post-stained with 1% aqueous uranyl acetate and Reynold’s lead citrate. All imaging was performed at 80 KV in a Talos 200X transmission electron microscope (Thermo Fisher Scientific, Hillsboro, OR, USA).

For SEM, the Mϕ cultures were fixed with 2.5% glutaraldehyde/1% paraformaldehyde in sodium cacodylate buffer, followed by 1% OsO_4_, then dehydrated through an ethyl alcohol series. The coverslips were critical-point-dried and coated with 2 nm iridium. The cells were imaged using a Teneo Scanning Electron Microscope (Thermo Fisher Scientific).

### 2.8. Statistical Analyses

Statistical analyses were performed using GraphPad Prism 7.0 software. Data sets were assessed by one-way ANOVAs followed by post hoc Tukey’s tests for multiple comparisons, and *p*-values < 0.05 were deemed as statistically significant.

## 3. Results

### 3.1. X. laevis IL34-Mϕs, but Not CSF1-Mϕs, Confer Anti-Bd Resistance In Vivo

Here, we explored the roles of *X. laevis* CSF1- and IL-34-Mϕs in anti-Bd defenses. To this end, we enriched these subsets in *X. laevis* skins by subcutaneous rCSF-1 and rIL34 injections. Compared to the animals injected with the rctrl, the subcutaneous administration of rCSF1 and rIL34 resulted in significant enrichment of esterase-positive Mϕs in the epidermal *X. laevis* skin (Figure 1A–D). Following challenge with Bd, the frogs with enriched IL34-Mϕs possessed significantly lower fungal loads, whereas the animals enriched with CSF1-Mϕs had skin Bd loads comparable to those seen in the rctrl-administered animals (Figure 1E). Irrespective of treatment, the skins of all the Bd-infected animals exhibited hallmark epidermal thickening, often accompanied by the presence of discharged zoosporangia (Figure 1F).

### 3.2. Both CSF1- and IL34-Mϕs Phagocytose Bd In Vitro

To study the biology of amphibian Mϕs, we previously optimized methods for generating *X. laevis* CSF1- and IL34-Mϕ cultures in vitro from bone marrow-derived Mϕ precursors [58]. Presently, we used these cultures to discern how the frog Mϕ cultures respond to Bd. Our SEM and TEM analyses of the *X. laevis* CSF1- and IL34-Mϕs corroborated our previous cytological comparisons [60], marking a striking difference in the morphologies of these two subsets (Figure 2A,B,F,G). Whereas the CSF1-Mϕs appeared larger with substantially more ruffled membranes, the IL34-Mϕs were relatively smaller and possessed many dendritic cell (DC)-like projections, consistent with their close relationship to frog DCs [61]. When we co-incubated CSF1- and IL34-Mϕs with Bd, we found that both Mϕ populations were equally capable of ingesting and destroying Bd zoospores and zoosporangia (CSF1-Mϕs: Figure 2C–E; IL34-Mϕs: Figure 2H–J).

### 3.3. CSF1- and IL34-Mϕs Cocultured with Bd Exhibit Disparate Polarization States

We next examined how Bd challenge affects the CSF1- and IL34-Mϕ expression of hallmark immune genes. Bd did not significantly alter the expression of NADPH oxidase catalytic domain genes (*noxa2*, *nox2*) in either Mϕ subset but resulted in significantly increased inducible nitric oxide synthase (*inos*, catalyzes reactive nitrogen production) gene expression in both Mϕ subtypes (Figure 3). Notably, Bd elicited significantly increased expression of the arginase-1 (*arg1*) and indoleamine 2,3 dioxygenase (*ido*) genes, which are archetypal of immunosuppressive Mϕs [62], in CSF1-Mϕs but not IL34-Mϕs (Figure 3). Moreover, while Bd elicited increased expression of genes encoding the pro-inflammatory tumor necrosis factor (*tnf*) and the immunosuppressive interleukin-10 (*il10*) cytokines in both Mϕ types, IL34-Mϕs had significantly greater transcript levels for *tnf*, while CSF1-Mϕs possessed significantly greater mRNA levels of *il10* (Figure 3).

### 3.4. Bd Gene Expression Differs When in Coculture with CSF1- or IL34-Mϕs

Bd cells produce a plethora of immunosuppressive compounds. Chiefly among these are kynurenine [63], culminating from indoleamine 2,3 dioxygenase (IDO)-mediated tryptophan breakdown [64], as well as methylthioadenosine (MTA) [62] and spermidine [65], which are a generated during ornithine breakdown by ornithine decarboxylase (ODC) and by spermidine synthase [66]. Interestingly, Bd co-incubated with IL34-Mϕs exhibited significantly decreased expression of its *ido* gene, suggesting less production of inhibitory kynurenines. Bd cocultured with CSF1-Mϕs possessed significantly increased expression of the Bd arginase-encoding *arg* gene, potentially leading to greater production of ornithine and inhibitory MTA and spermidine. We did not see significant changes in the expression levels of spermidine synthase (*srm*; Figure 4). Bd cocultured with CSF1-Mϕs, but not IL34-Mϕs, also possessed a greater expression of antioxidant enzyme genes, catalase (*cat*), and superoxide dismutase (*sdm*; Figure 4).

## 4. Discussion

Tissue-resident Mϕ-lineage cells serve as critical immune sentinels during fungal infections. They recognize fungal (among other) pathogens via a number of innate immune receptors, initiate antifungal responses, and directly eliminate the invading fungi through mechanisms such as phagocytosis and reactive radical-mediated clearance [67]. The CSF1R ligands, CSF1 and IL34 are important to the development of most if not all Mϕ subsets [51]. With respect to cutaneous immunity, IL34 appears to be more critical to the maintenance of mammalian Langerhans cells under steady-state conditions [68], whereas CSF1 is believed to contribute to the differentiation of Langerhans cells during skin inflammatory responses [69], as reviewed in [52,53]. Langerhans-like cells have been described in several amphibian species [48,70,71]. These cells possess ATPase activity, express MHC class II, and stain positive with mammalian anti-langerin antibodies (CD207; hallmark marker of mammalian Langerhans cells [72]). Whether these cells are functional equivalents to mammalian Langerhans cells and what their roles are during Bd infections remains unexplored. While we know that some of the functional dichotomy between these frog Mϕ subsets has been evolutionarily conserved, we also anticipate that the unique amphibian physiology will dictate key differences between frog CSF1- and IL34-Mϕ subsets and their mammalian counterparts. The subcutaneous administration of rCSF1 and rIL34 resulted in the enrichment of non-specific esterase-positive cells (marker of Mϕ-lineage) in frog epidermises, while in vitro-derived IL34-Mϕs resembled mammalian dendritic cells, suggesting that as in mammals [68], the frog IL34 may play a role in the development of Langerhans-like cells at steady-state. Possibly, also akin to mammals [69], the frog CSF1 may be involved in generating Langerhans-like Mϕs during inflammatory responses. It is worth mentioning that while the mammalian-specific anti-langerin antibody recognizes cells in amphibian skins [71], to our knowledge the gene encoding the amphibian version of this protein has not been identified. Moreover, other mammalian myeloid cells besides Langerhans cells express langerin [73,74,75]. As such, we refrain from calling the frog epidermal skin CSF1- or IL34-Mϕs “Langerhans cells”, but in accordance with the findings presented here, we propose that these cell subsets play disparate roles during Bd infections of frog skin.

The continuum of Mϕ functional polarization is defined by the pro-inflammatory (M1) and the immunosuppressive (M2) polar extremes [62]. Our past work suggests that *X. laevis* CSF1-Mϕs may be more inflammatory than IL34-Mϕs in some contexts [55,76]. Moreover, there is a growing body of literature indicating that in mammals, IL34 is produced in tolerogenic contexts by cells such as T regulatory (Treg) cells [77] while mammalian IL34-Mϕs adopt immunosuppressive phenotypes (reviewed in [53]). Conversely, our present work indicates that Bd coculture results in CSF1-Mϕs adopting an M2-like transcriptional profile, including upregulated expression of hallmark M2 genes such as arginase-1 and interleukin-10. CSF1-Mϕs cocultured with Bd also possessed increased gene expression of the indoleamine 2,3 dioxygenase (IDO) enzyme, which catalyzes tryptophan breakdown, the byproducts of which inhibit lymphocytes [78]. By contrast, Bd-challenged IL34-Mϕs possessed significantly greater expression of the pro-inflammatory tumor necrosis factor cytokine, the mammalian counterpart of which is prominently involved in antifungal defenses [79]. There is the possibility that our in vitro Mϕ cultures do not reflect what happens to CSF1- and IL34-Mϕs in vivo during Bd infections. However, our previous studies do indicate that these in vitro-derived CSF1- and IL34-Mϕ cultures recapitulate well the respective in vivo Mϕ subset responses to viral and mycobacterial pathogens [55,56,58,80]. Moreover, the more thoroughly characterized mammalian CSF1- and IL34-Mϕ polarization states are likewise context-dependent, and their respective pro- and anti-inflammatory phenotypes may be altered by additional immune stimuli (reviewed in [53]). Future time courses and ex vivo studies of these Mϕ subsets in the context of Bd infections will provide greater clarity for how the frog CSF1- and IL34-Mϕs relate to their mammalian counterparts and how their activation states contribute to chytrid infection outcomes.

Phagocytosis and reactive oxygen responses are major mechanisms by which mammalian Mϕs eliminate pathogenic fungi [67]. Both CSF1- and IL34-Mϕs phagocytosed Bd zoosporangia in vitro, although only IL34-Mϕs protected animals during in vivo infections. *X. laevis* IL34-Mϕs possess significantly more robust reactive oxygen responses than CSF1-Mϕs [56,58]. Coculture with Bd did not alter the IL34-Mϕs gene expression of catalytic domains (*noxa2*, *nox2*) of the NADPH oxidase complex responsible for ROS production. Conversely, following Bd-coculture, both frog Mϕ subsets upregulated their gene expression of iNOS, which catalyzes reactive nitrogen intermediate (RNI) production. While reactive oxygen responses are more prominently associated with Mϕ-mediated clearance of fungal pathogens, there are some contexts in which reactive nitrogen responses contribute to Mϕ antifungal responses [67]. Our *X. laevis* Mϕ model will be useful in future in vitro, ex vivo, and in vivo studies to discern the extent to which Mϕ ROI, RNI, and other responses contribute to Bd clearance.

Akin to other fungal pathogens, Bd possesses many immune evasion strategies [81]. Amongst these, Bd produces immunosuppressive metabolites such as kynurenine, methylthioadenosine, and spermidine [63,65], catalyzed by the Bd-encoded indoleamine 2,3 dioxygenase, arginase, and ornithine decarboxylase enzymes, respectively. Bd also encodes superoxide dismutase and catalase enzymes, which protect it from myeloid cell-derived reactive oxygen species. Bd co-incubated with CSF1-Mϕs, but not with IL34-Mϕs, upregulated its arginase, superoxide dismutase, and catalase gene expression. Conversely, Bd cocultured with IL34-Mϕs, but not with CSF1-Mϕs, downregulated its *ido* gene expression. Presumably, Bd-CSF1-Mϕ interactions facilitate fungal immune evasion and possibly manifest in exacerbated infections at later times to those examined here. This Bd-CSF1-Mϕ immune suppression is not surprising, as our previous work indicates that this frog Mϕ subset is likewise prone to immune suppression by the Frog Virus 3 ranavirus [80] and *Mycobactrerium marinum* [80]. On the other hand, the in vitro findings that IL34-Mϕs are relatively less susceptible to Bd immune suppression and adopt more inflammatory phenotypes are consistent with our observation that enriching these cells in vivo results in greater anti-Bd protection. It is difficult to speculate what cues alert Bd to upregulate its expression of immunomodulatory genes when in CSF1-Mϕ but not IL34-Mϕ coculture. The observed differences in Bd gene expression in response to the two Mϕ subsets suggest that this fungus possesses mechanisms for sensing and adapting to distinct leukocyte subtypes. This notion may prove to be useful in future efforts to prevent chytridiomycosis and the spread of this fungal pathogen.

## 5. Conclusions

Mammalian Mϕ-lineage cells represent a critical line of defense against fungal pathogens [67]. Through the present work, we show that the same is true of the *X. laevis* frog IL34-Mϕs but not CSF1-Mϕs. Further studies of amphibian CSF1- and IL34-Mϕs will undoubtedly provide new perspectives on the immune cell contributions to Bd susceptibility and resistance, as well as the evolutionarily diverged and converging pathways of Mϕ antifungal immunity.

## Figures and Tables

**Figure 1 jof-11-00311-f001:**
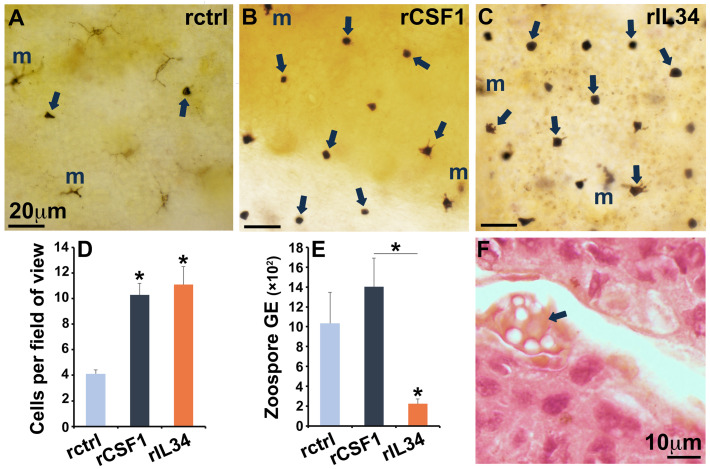
Enrichment of cutaneous CSF1- and IL34-Mϕs and Bd challenge. Non-specific esterase (NSE) stain of epidermal skin layers from the (**A**) rctrl- and (**B**) rCSF1- and (**C**) rIL34-administered frogs. Arrows indicate myeloid (Mϕs) cells, and melanocytes are denoted by ‘m’. Images are representative of 4 animals per treatment group (*N* = 4). (**D**) Means ± SEMs of the number of NSE-positive cells per field of view of skin from frogs 3 days after subcutaneous administration of the rctrl or rCSF1 or rIL34 (*N* = 4 frogs/treatment group). (**E**) Bd loads (zoospore genomic equivalents, GE) in dorsal skins of frogs 3 days after control (rctrl)-, CSF1-, and IL34-Mϕ-enrichment followed by 7 days of Bd challenge (*N* = 7 frogs/treatment group). (**F**) H&E stain of Bd (arrow)-infected skin. Asterisks above bars indicate statistical significance from the rctrl groups, and asterisks above lines are indicative of statistical differences between the treatment groups denoted by the lines, *p* < 0.05.

**Figure 2 jof-11-00311-f002:**
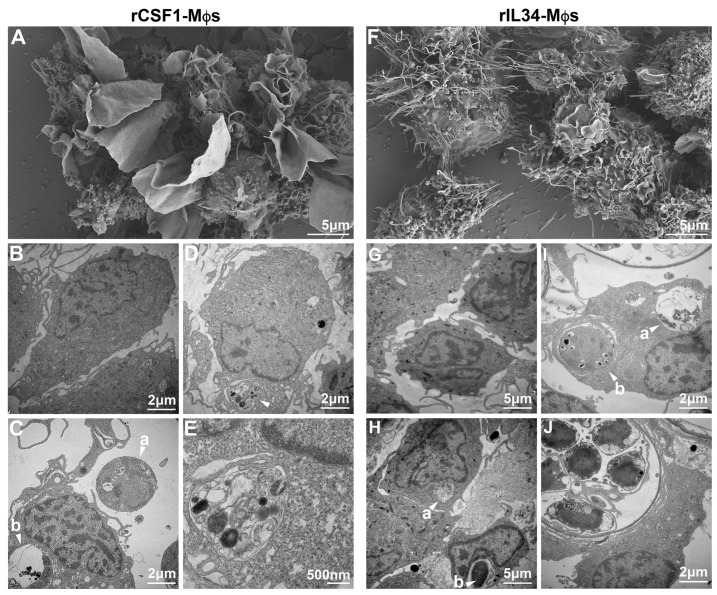
Scanning and transmission electron microscopy images of (**A**–**E**) CSF1- and (**F**–**J**) IL34-Mϕs. CSF1- and IL34-Mϕs were incubated alone (**A**,**B** and **F**,**G**, respectively) or with developing Bd thalli (**C**–**E** and **H**–**J**, respectively) for 5 h (5 Bd/Mϕ) before preparation and electron microscopy analyses. (**A**) SEM of CSF1-Mϕs. (**B**) TEM of CSF1-Mϕs. (**C**) TEM of CSF1-Mϕs co-incubated with Bd. Arrows: (a) non-phagocytosed zoosporangium; (b) phagocytosed and degraded zoosporangium. (**D**) TEM of CSF1-Mϕs co-incubated with Bd. Arrows: phagocytosed zoosporangium. (**E**) Higher magnification of zoosporangium-containing phagosome in (**D**). (**F**) SEM of IL34-Mϕs. (**G**) TEM of IL34-Mϕs. (**H**) TEM of IL34-Mϕs co-incubated with Bd. Arrows: (a) phagocytosed Bd zoospore in the process of being degraded; (b) phagocytosed zoospore. (**I**) TEM of IL34-Mϕs co-incubated with Bd. Arrows: (a) phagocytosed, degraded zoosporangium; (b) phagocytosed zoosporangium. (**J**) Higher magnification of mature zoosporangium-containing phagosome.

**Figure 3 jof-11-00311-f003:**
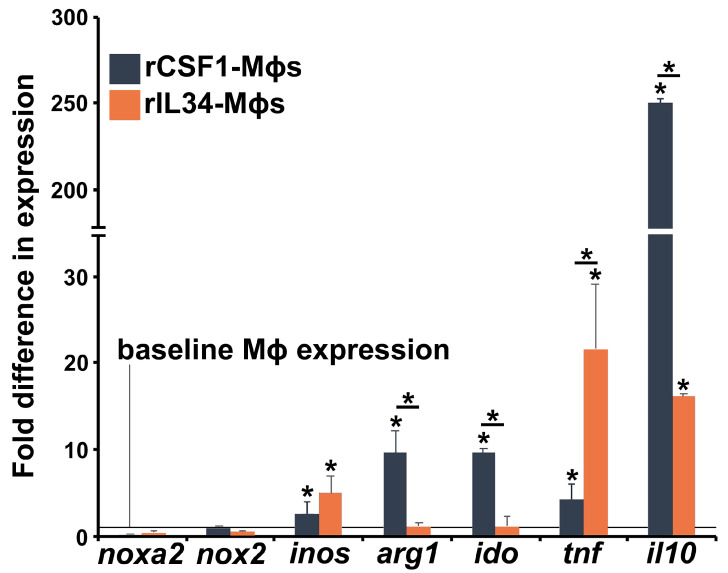
Consequences of Bd exposure on CSF1- and IL34-Mϕ activation. CSF1- and IL34-Mϕs were incubated with developing Bd thalli for 5 h (5 Bd/Mϕ), and the cocultures were examined for their expression of immune genes relative to *gapdh* endogenous control. All data are represented relative to parallel respective CSF1- and IL34-Mϕ cultures, incubated without Bd, depicted by the horizontal line. The results are means ± SEMs of immune gene expression from 6 independent CSF1- and IL34-Mϕ cultures, derived from 6 individual frogs, *N* = 6. Asterisks above bars indicate statistical significance between baseline and Bd-elicited Mϕ gene expression, and asterisks above lines are indicative of statistical differences between the treatment groups denoted by the lines, *p* < 0.05. The examined frog Mϕ genes included *noxa2* and *nox2*-NADPH oxidase catalytic domains; *inos*—inducible nitric oxide synthase; *arg1*—arginase-1; *ido*—indoleamine 2,3 dioxygenase; *tnf*—tumor necrosis factor, and *il10*—interleukin-10.

**Figure 4 jof-11-00311-f004:**
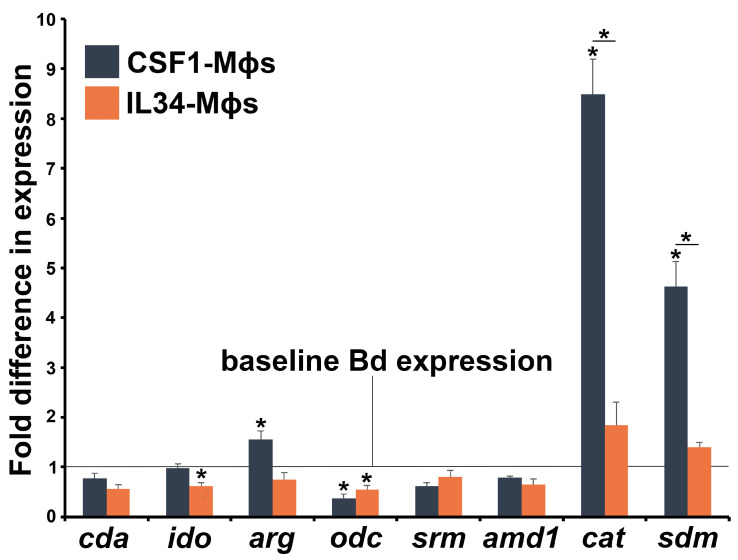
Consequences of Mϕ-Bd coculture on Bd gene expression. Developing Bd thalli were incubated alone or in coculture with CSF1- and IL34-Mϕs for 5 h (5 Bd/Mϕ), and Bd was examined for expression of immune evasion genes relative to the Bd *gapdh* endogenous control. All data are represented relative to baseline Bd gene expression, depicted by the horizontal line. The results are means ± SEMs of Bd gene expression from 6 independent CSF1- and IL34-Mϕ cocultures, with frog cells derived from 6 individual frogs, *N* = 6. Asterisks above bars indicate statistical significance between baseline and Mϕ-elicited Bd gene expression, and asterisks above lines are indicative of statistical differences between the treatment groups denoted by the lines, *p* < 0.05. The examined Bd genes included *arg*—arginase; *amd*—adenosylmethionine decarboxylase; *cat*—catalase; *cda*—chitin deacetylase; *ido*—indoleamine 2,3 dioxygenase; *odc*—ornithine decarboxylase; *sdm*—superoxide dismutase; and *srm*—spermidine synthase.

## Data Availability

The original contributions presented in this study are included in the article/Appendix A. Further inquiries can be directed to the corresponding author.

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
