# Peer review of "Amphibian (Xenopus laevis) Macrophage Subsets Vary in Their Responses to the Chytrid Fungus Batrachochytrium dendrobatidis"

_jof, 2025, doi:10.3390/jof11040311_

Round 1
Reviewer 1 Report
This study, which is of particular pertinence, describes the responses of two distinct amphibian macrophage subsets to the chtrid fungus Batrachochytrium dendrobatidis. The publication is illustrated with clarity and provides novel insights into the contributions of immune cells to susceptibility to Bd and antifungal immunity.
Minor remarks:
Line 78: why are antibiotics supplemented to the culture medium?
Line 116: please provide some more details on the protocol ‘optimized to frog tissues’
Author Response
Reviewer 1:
This study, which is of particular pertinence, describes the responses of two distinct amphibian macrophage subsets to the chtrid fungus Batrachochytrium dendrobatidis. The publication is illustrated with clarity and provides novel insights into the contributions of immune cells to susceptibility to Bd and antifungal immunity.
We thank the reviewer for their kind words
Minor remarks:
Line 78: why are antibiotics supplemented to the culture medium?
To prevent potential growth of other microbes.
Line 116: please provide some more details on the protocol ‘optimized to frog tissues’
We only strayed from the manufacturer’s protocol by diluting the hematoxylin solution 1/3 in APBS, as is not stated in our revised manuscript.

Reviewer 2 Report
This manuscript presents a valuable and well-structured contribution to the field of amphibian immunology, offering novel insights into how distinct macrophage subsets modulate host responses to Batrachochytrium dendrobatidis. The research is timely and of high relevance given the continued global impact of chytridiomycosis on amphibian populations. By using Xenopus laevis as a model, combined with both in vivo and in vitro approaches, the authors provide strong evidence for the functional divergence between CSF1- and IL34-derived macrophages in antifungal defense. The insights into Bd’s differential gene expression in response to these macrophage subtypes are particularly compelling. However, to further enhance the scientific clarity and broader relevance of the work, the authors are encouraged to: (1) clarifying the novelty of their approach in light of existing literature; (2) incorporate recent, comprehensive reviews such as Muñoz-Garcia et al. (2021) and Freuchet et al. (2021) that highlight IL-34’s context-specific immunological roles; and (3) refine the Discussion by reducing repetition, especially where macrophage polarization and Bd immune evasion mechanisms are described.
This study provides valuable insights into macrophage roles in amphibian defense against Batrachochytrium dendrobatidis (Bd). The following suggestions aim to improve clarity, scientific rigor, and contextual depth:
- Abstract (Lines 11–27):
- Necessary: Replace “Mφ” with “macrophages” to avoid potential rendering issues and improve accessibility.
- Desirable: Expand the concluding sentence to better frame the broader impact or future applications.
- Keywords (Line 28):
- Desirable: Refine for specificity—replace “chytrid” with “chytridiomycosis,” and consider adding terms like “amphibian immune response,” “CSF1,” and “IL34” to better reflect the molecular and ecological context.
- Introduction:
- Necessary: Revise the final paragraph (Lines 63–68) to state the study objective and general approach rather than summarizing results. I suggest replacing it with: "In this study, we investigated the distinct roles of CSF1- and IL34-differentiated macrophages in Xenopus laevis during Bd infection, using both in vivo and in vitro approaches to assess their contribution to antifungal defense and host–pathogen interactions”. This revision preserves the purpose of the study while adhering to standard scientific writing conventions.
- Methods (Lines 69–190):
- Necessary: Clarify in Line 116 whether macrophage enrichment was assessed qualitatively or quantitatively (e.g., cell counts, image analysis).
- Desirable: In Lines 110–111 and 118–119, indicate whether the cytokine dose (5 µg/animal), timing (3-day interval), and Bd-to-macrophage ratio (5:1) were based on prior data or optimization.
- Necessary: In Line 118, justify the sample sizes (N=4 and N=7) by referencing statistical power or prior studies.
- Necessary: In Lines 132–134, clarify how macrophage differentiation was confirmed (morphological or molecular markers), or cite previous validation.
- Electron Microscopy (Lines 169–185):
- Desirable: Specify what cellular structures or interactions were examined via SEM/TEM (e.g., phagocytosis, membrane changes) to contextualize the rationale for using EM.
- Results (Lines 193–284):
- Desirable: Rephrase the section title “Both CSF1- and IL34-Mφs phagocytose Bd in vitro” to enhance clarity. Suggested options:
- “In Vitro Phagocytosis of Bd by CSF1- and IL34-Macrophages” or
- “CSF1- and IL34-Macrophages Exhibit Comparable Bd Phagocytosis In Vitro”
- Desirable: If available, add quantitative data on phagocytic activity (e.g., % of macrophages with ingested Bd) to support the conclusion of equal capacity.
- Discussion (Lines 325–427):
- Necessary: Avoid repetition—consolidate recurring points about polarization differences (e.g., Lines 341–344 vs. 395–397).
- Desirable: Expand the discussion of Bd’s differential gene expression in response to macrophage types. Emphasize the implication that Bd may actively sense and adapt to specific immune phenotypes.
- References:
- Necessary: Include the following two recent and relevant reviews to strengthen your background and discussion:
- Muñoz-Garcia, J., et al. (2021). The twin cytokines interleukin-34 and CSF-1: masterful conductors of macrophage homeostasis. Theranostics, 11(4), 1568.
- Freuchet, A., et al. (2021). IL-34 and CSF-1, deciphering similarities and differences at steady state and in diseases. Journal of Leukocyte Biology, 110(4), 771–796.
These references provide up-to-date insights into IL-34’s role in tissue-specific macrophage function and immune modulation and are not currently cited in your reference list.
Author Response
This manuscript presents a valuable and well-structured contribution to the field of amphibian immunology, offering novel insights into how distinct macrophage subsets modulate host responses to Batrachochytrium dendrobatidis. The research is timely and of high relevance given the continued global impact of chytridiomycosis on amphibian populations. By using Xenopus laevis as a model, combined with both in vivo and in vitro approaches, the authors provide strong evidence for the functional divergence between CSF1- and IL34-derived macrophages in antifungal defense. The insights into Bd’s differential gene expression in response to these macrophage subtypes are particularly compelling.
However, to further enhance the scientific clarity and broader relevance of the work, the authors are encouraged to: (1) clarifying the novelty of their approach in light of existing literature; (2) incorporate recent, comprehensive reviews such as Muñoz-Garcia et al. (2021) and Freuchet et al. (2021) that highlight IL-34’s context-specific immunological roles; and (3) refine the Discussion by reducing repetition, especially where macrophage polarization and Bd immune evasion mechanisms are described.
We thank the reviewer for this suggestion. Please find that we have substantially revised our discussion section, incorporating the suggested references and streamlining the content therein.
This study provides valuable insights into macrophage roles in amphibian defense against Batrachochytrium dendrobatidis (Bd). The following suggestions aim to improve clarity, scientific rigor, and contextual depth:
Abstract (Lines 11–27):
Necessary: Replace “Mφ” with “macrophages” to avoid potential rendering issues and improve accessibility.
Thank you and changed accordingly.
Desirable: Expand the concluding sentence to better frame the broader impact or future applications.
Thank you for the suggestion. Please find that we changed the last sentence of the abstract to “This work marks a step forward in our understanding of the roles of skin-resident leukocytes in antifungal defenses.”
Keywords (Line 28):
Desirable: Refine for specificity—replace “chytrid” with “chytridiomycosis,” and consider adding terms like “amphibian immune response,” “CSF1,” and “IL34” to better reflect the molecular and ecological context.
Please find that we have revised our manuscript accordingly
Introduction:
Necessary: Revise the final paragraph (Lines 63–68) to state the study objective and general approach rather than summarizing results.
I suggest replacing it with: "In this study, we investigated the distinct roles of CSF1- and IL34-differentiated macrophages in Xenopus laevis during Bd infection, using both in vivo and in vitro approaches to assess their contribution to antifungal defense and host–pathogen interactions”. This revision preserves the purpose of the study while adhering to standard scientific writing conventions.
Thank you for the suggestion. Please find that we changed the last paragraph of our introduction accordingly.
Methods (Lines 69–190):
Necessary: Clarify in Line 116 whether macrophage enrichment was assessed qualitatively or quantitatively (e.g., cell counts, image analysis).
Please find that we added the word ‘quantitatively’ to indicate as such.
Desirable: In Lines 110–111 and 118–119, indicate whether the cytokine dose (5 µg/animal), timing (3-day interval), and Bd-to-macrophage ratio (5:1) were based on prior data or optimization.
Thank you for the suggestions. Please find that they have been incorporated into the revised manuscript.
Necessary: In Line 118, justify the sample sizes (N=4 and N=7) by referencing statistical power or prior studies.
Thank you. Please note that we added wording: “based on prior experiments of similar nature”
Necessary: In Lines 132–134, clarify how macrophage differentiation was confirmed (morphological or molecular markers), or cite previous validation.
Thank you and please find that we cited previous work that characterized these cultures.
Electron Microscopy (Lines 169–185):
Desirable: Specify what cellular structures or interactions were examined via SEM/TEM (e.g., phagocytosis, membrane changes) to contextualize the rationale for using EM.
Thank you. Please find that we added a sentence saying that “Electron microscopy was performed to visualize differences between the two Mf subsets with respect to morphology and phagocytosis of Bd.”
Results (Lines 193–284):
Desirable: Rephrase the section title “Both CSF1- and IL34-Mφs phagocytose Bd in vitro” to enhance clarity. Suggested options:
“In Vitro Phagocytosis of Bd by CSF1- and IL34-Macrophages” or
“CSF1- and IL34-Macrophages Exhibit Comparable Bd Phagocytosis In Vitro”
Many thanks for the suggestion, which has been incorporated into our revised manuscript. The heading now reads “CSF1- and IL34-Mfs exhibit comparable phagocytosis of Bd in vitro”.
Desirable: If available, add quantitative data on phagocytic activity (e.g., % of macrophages with ingested Bd) to support the conclusion of equal capacity.
Unfortunately, we do not have quantitative data for this assay since it was difficult and costly to get many good images.
Discussion (Lines 325–427):
Necessary: Avoid repetition—consolidate recurring points about polarization differences (e.g., Lines 341–344 vs. 395–397)/
We thank the reviewer for pointing out this flaw in our manuscript. Please find that we have gone through and revised our discussion to be less redundant. We feel that this has substantially improved the flow and cohesiveness of our discussion.
Desirable: Expand the discussion of Bd’s differential gene expression in response to macrophage types. Emphasize the implication that Bd may actively sense and adapt to specific immune phenotypes.
Thank you for this suggestion. Please find that we added the following statement to our manuscript:
“In turn, the observed differences in Bd gene expression in response to the two Mf subsets suggests that this fungus possesses mechanisms for sensing and adapting to distinct leukocyte subtypes.”
References:
Necessary: Include the following two recent and relevant reviews to strengthen your background and discussion:
- Muñoz-Garcia, J., et al. (2021). The twin cytokines interleukin-34 and CSF-1: masterful conductors of macrophage homeostasis. Theranostics, 11(4), 1568.
- Freuchet, A., et al. (2021). IL-34 and CSF-1, deciphering similarities and differences at steady state and in diseases. Journal of Leukocyte Biology, 110(4), 771–796.
These references provide up-to-date insights into IL-34’s role in tissue-specific macrophage function and immune modulation and are not currently cited in your reference list.
We thank the reviewer for this excellent suggestion. Please find that we have added these references and cited them, where appropriate, in our revised manuscript.

Reviewer 3 Report
A solid study clearly demonstrating that two different subset of frog macrophages obtained by treatments with recombinant frog interleukin 34 or colony stimulating factor-1 respectively differ in their response to an important chytrid pathogen. This is shown by a combination of experiments in vivo, in vitro with isolated leukocytes and co-cultivating macrophages and the chytrid fungus. The IL-34 treated macrophages were convincingly shown to be more effective in reducing the fungal infection compared to the CSF1 treated macrophages in spite of being equally effective in phagocytizing fungal cells. The different immune gene transcript profiles obtained from the two macrophage subsets after exposure to the fungus may hold important clues to the mechanisms involved in the anti-fungal response by the host and constitute a foundation for further research into these mechanisms.
No changes required.
Author Response
A solid study clearly demonstrating that two different subset of frog macrophages obtained by treatments with recombinant frog interleukin 34 or colony stimulating factor-1 respectively differ in their response to an important chytrid pathogen. This is shown by a combination of experiments in vivo, in vitro with isolated leukocytes and co-cultivating macrophages and the chytrid fungus. The IL-34 treated macrophages were convincingly shown to be more effective in reducing the fungal infection compared to the CSF1 treated macrophages in spite of being equally effective in phagocytizing fungal cells. The different immune gene transcript profiles obtained from the two macrophage subsets after exposure to the fungus may hold important clues to the mechanisms involved in the anti-fungal response by the host and constitute a foundation for further research into these mechanisms.
No changes required.
Many thanks for your kind words.

Reviewer 4 Report
The research work is conducted with a sound approach, and the experiments performed provide results that support the discussion and conclusions reached. However, there are suggestions for the manuscript.
First, do CSF1 and IL34 in Xenopus laevis possess post-translational modifications? And that both recombinants do not have any type of modification, or perhaps they are different. This is a relevant factor for macrophage activation due to CSF1 and not IL34.
It's worth noting these aspects if you already have that information.
It could also be that removing the IL34 signal peptide affects this signaling. Also, if you already have that information, you can use it as a backup.
Second, regarding lines 83-106, I believe it's important to show these results, at least in the supplementary material. For the recombinant products used in macrophage activation in coculture, it's essential to ensure that the peptides are of good quality and that no components interfere with activation.
ml --> mL
hr --> h
Line: 120. Extraction DNA o RNA?
Author Response
Reviewer 4:
The research work is conducted with a sound approach, and the experiments performed provide results that support the discussion and conclusions reached. However, there are suggestions for the manuscript.
We thank the reviewer for their kind words and suggestions.
First, do CSF1 and IL34 in Xenopus laevis possess post-translational modifications? And that both recombinants do not have any type of modification, or perhaps they are different. This is a relevant factor for macrophage activation due to CSF1 and not IL34. It's worth noting these aspects if you already have that information.
Please note that the recombinant CSF1 and IL34 were generated using a eukaryotic expression system (Sf9 insect cells), which glycosylate and fold the proteins. While we cannot guarantee that the glycosylation patterns are entirely the same as what we would expect from frog CSF1/IL34 producing cells, these proteins would be properly folded and glycosylated.
It could also be that removing the IL34 signal peptide affects this signaling. Also, if you already have that information, you can use it as a backup.
To our knowledge, the mature secreted IL34 lacks as signal peptide. Thus, the recombinant form of this cytokine was engineered to represent the signal peptide-cleaved version of the protein. We do not investigate CSF1R signaling in the present manuscript.
Second, regarding lines 83-106, I believe it's important to show these results, at least in the supplementary material. For the recombinant products used in macrophage activation in coculture, it's essential to ensure that the peptides are of good quality and that no components interfere with activation.
Thank you. Please find that we added a supplemental figure, showing rCSF1 and rIL34 fractions and concentrated protein.
ml --> mL
hr --> h
Thank you. To our knowledge, both are acceptable formats.
Line: 120. Extraction DNA o RNA?
Extraction of DNA, as indicated.